# Negligible Effects of Nutraceuticals from Beetroot Extract on Cardiovascular and Autonomic Recovery Response following Submaximal Aerobic Exercise in Physically Active Healthy Males: A Randomized Trial

**DOI:** 10.3390/ijerph20054019

**Published:** 2023-02-23

**Authors:** Cicero Jonas R. Benjamim, Francisco Welington de Sousa Júnior, Andrey Alves Porto, Camila Venancia Guerra Andrade, Maria Íris L. Saraiva de Figueiredo, Cicera Josilânia R. Benjamim, Guilherme da Silva Rodrigues, Elida M. Braga Rocha, Taisy Ferro Cavalcante, David M. Garner, Vitor Engracia Valenti, Carlos R. Bueno Júnior

**Affiliations:** 1Department of Internal Medicine, Ribeirão Preto Medical School, University of São Paulo, Ribeirão Preto 14049-900, SP, Brazil; 2University Center of the Juazeiro do Norte, Juazeiro do Norte 63010-475, CE, Brazil; 3 São Paulo State University (UNESP), Presidente Prudente, São Paulo Brazil; Autonomic Nervous System Center, UNESP, Marília 17525-900, São Paulo; 4Development Nutrition, Phytotherapy and Hygiene Research Group, University of Pernambuco, Petrolina 56328-900, PE, Brazil; 5Cardiorespiratory Research Group, Department of Biological and Medical Sciences, Faculty of Health and Life Sciences, Oxford Brookes University, Oxford OX3 0BP, UK

**Keywords:** *Beta vulgaris* L., dietary supplements, exercise, autonomic nervous system, cardiovascular system

## Abstract

Background: There is little evidence that nutraceuticals from beetroot extract are beneficial with regards to recovery of the cardiovascular parameters and the autonomic nervous system (ANS) after submaximal aerobic exercise, though this formulation is employed widely for this purpose. Objective: To study the effects of beetroot extract supplementation on the recovery of cardiorespiratory and autonomic parameters after a session of submaximal aerobic exercise. Methods: Sixteen healthy male adults commenced a cross-over, randomized, double-blind and placebo-controlled trial. Beetroot extract (600 mg) or placebo (600 mg) were ingested 120 min prior to evaluation on randomized days. We assessed systolic blood pressure (SBP), diastolic blood pressure (DBP), pulse pressure (PP), mean arterial pressure (MAP), heart rate (HR) and HR variability (HRV) indexes at Rest and during 60 min of recovery from submaximal aerobic exercise. Results: Beetroot extract ingestion slightly accelerated HR, SBP, DBP and MAP reduction following exercise associated to the placebo protocol (vs. rest). Yet no group effect (*p* = 0.99) was identified between the beetroot and placebo protocols on HR mean, in addition to interaction (group vs. time) (*p* = 0.60). No group effect was attained between the SBP (*p* = 0.90), DBP (*p* = 0.88), MAP (*p* = 0.73) and PP (*p* = 0.99) protocols and no significant differences (group vs. time) were observed between the values of SBP (*p* = 0.75), DBP (*p* = 0.79), MAP (*p* = 0.93) and PP (*p* = 0.63) between placebo and beetroot protocols. Similarly, the reoccurrence of cardiac vagal modulation after exercise via the HF (ms^2^) was enhanced, but not with regards to the RMSSD index. No group effect (*p* = 0.99) was identified for the HF (*p* = 0.90) and RMSSD (*p* = 0.67) indices. Likewise, we observed no significant differences (group vs. time) amongst the values of HF (*p* = 0.69) and RMSSD (*p* = 0.95) between the placebo and beetroot protocols. Conclusion: Whilst beetroot extract might assist the recovery of the cardiovascular and autonomic systems following submaximal aerobic exercise in healthy males, these results seem insignificant owing to minor differences between interventions and are weak clinically.

## 1. Introduction

Amongst the compounds in beetroot (*Beta vulgaris* L.), betaine and nitrate (NO_3_) are the most widely recognized [1,2]. These two compounds have shown promising results with regards to cardiovascular activity [1,2] and are able to optimize cardiovascular parameters (e.g., blood pressure and heart rate), both in healthy populations, above and beyond the prevalence of cardiovascular diseases (e.g., hypertension, coronary heart disease) [3,4].

Much consideration has been given to the effects of nitrate. Consequently, the effects of other beet compounds are repeatedly unnoticed. Despite this, the use of isolated NO_3_ in the form of salts (e.g., sodium and potassium NO_3_) and not from vegetable origins is often questioned, as the entirety of its effects seem to be dependent on an interaction with other compounds originating in vegetables. This has been established in both the results of studies of cardiovascular and sports performances. A recent systematic review study with meta-analysis verified that beetroot juice rich in NO_3_ was capable of decreasing systolic blood pressure by −4.95 mmHg in patients with arterial hypertension [5]. In another systematic review, when NO_3_ was enforced in the form of salts, blood pressure reduction in the same population was absent [6]. When we investigate performance, these identical effects occur; in the meta-analysis by Silva et al. [7], the use of NO_3_ salts was inefficient at improving physical performance if it originated from a diet supplemented with vegetables or beet juice.

Furthermore, the consumption of nutraceuticals (e.g., betaine, anthocyanins, flavonoids) has now revealed vital direct and indirect contributions to cardiovascular health. Betaine is an anti-inflammatory agent that has been revealed to be effective in lessening homocysteine levels, based on an amino acid which when elevated in the bloodstream becomes a cardiovascular disease risk factor [8,9]. Anthocyanins and flavonoid activity have inflammatory activity in addition to their cardioprotective effects, for instance, a reduction in blood pressure and upgrading of endothelial, vascular and neuronal function [10]. In this way, some studies have specified that these compounds in beetroot can interact and optimize cardiac autonomic control [11,12]. Cardiac activity is somewhat controlled by the autonomic nervous system (ANS) and its sympathetic and parasympathetic branches. Through the intervals between heartbeats (RR intervals) or heart rate variability (HRV), it is plausible to assess cardiac autonomic modulation [13]. Healthy individuals show prominent parasympathetic modulation and greater variability between RR intervals. In these situations, there is an enhanced transition capacity in the ANS response (vagal return) to the heart after such challenging situations (e.g., physical exercise), triggering a sudden reduction in HR [14]. In the past few years, scientific publications have frequently applied the RMSSD (root mean square standard deviation) and HF (high frequency) indexes as a way to assess the cardiac vagal modulation recovery in response to physical exercise. These two indexes have been revealed to be fairly accurate in comprehending the cardiac autonomic modulation. The RMSSD index is highly representative of the parasympathetic modulation of the heart and the HF index experiences the negative influence of respiratory frequency [13].

Analysis of cardiac activity recovery following exercise is a practical and reliable method for stratifying the risk of disease and antagonistic cardiovascular events [13]. Several studies have focused on the investigation of nutritional interventions (e.g., energy drinks and caffeine) [15,16,17,18] that can induce a delay in the recovery of HR and HRV and thus increase cardiovascular risks. Other studies have pursued the identification of compounds that maximize these parameters and minimalize cardiac stress caused by physical exercise [19].

Preliminary evidence exposed that beetroot extract can accelerate HR and HRV recoveries by encouraging a rapid return of parasympathetic modulation to the heart after performing high-intensity exercise [19]. Despite this, nothing confirms that beetroot extract can improve the cardiovascular system in response to submaximal aerobic exercise. Corresponding investigations are critical to heighten the external validity of previous studies’ findings. Bearing in mind that aerobic exercises (e.g., walking, running) are more prevalent in the general populations’ routine [20], it is vital to study if beetroot compounds can optimize the recovery of cardiac activity after this type of physical exercise. Moreover, we elected to test the beetroot extract in preference to beetroot juice because its taste is unpalatable to many individuals and the extract can be a different way of providing the beetroot compounds. This study anticipated an inspection of the effects of beetroot extract on the recovery of cardiovascular and autonomic parameters following a session of submaximal aerobic exercise. Our theory is that beetroot extract ingestion has no effects on cardiovascular and autonomic system in recovery to submaximal aerobic exercise.

## 2. Materials and Methods

This is a cross-over study, randomized, double-blind and placebo-controlled registered in Clinicaltrials.gov under protocol number NCT05294198 (https://clinicaltrials.gov/ct2/show/NCT05294198 (accessed on 10 December 2022)) and reported following the Consolidated Standards of Reporting Trials (CONSORT) statement [21]. University of Pernambuco Research Ethics Committee approved the procedures of this trial (Number: 22562719.5.0000.5191—21 October 2019), consistent with the most recent Helsinki Declaration.

### 2.1. Participants

We screened 21 healthy college-educated males. Inclusion criteria were defined as young males (18 to 30 years old), physically active according to International Physical Activity Questionnaire (IPAQ) [22] and body mass index (BMI) between >18.5 kg/m^2^ and <29.9 kg/m^2^. Smokers, subjects undergoing pharmacotherapies with musculoskeletal, metabolic, renal diseases and those unable to complete the necessary stages of the experimental protocol were the exclusion criteria (Figure 1). In the final event, only 16 males accomplished all of the protocols.

### 2.2. Initial Assessment

The subjects were documented by data assemblage, for instance, age (years), mass (kg), height (cm), heart rate (beats-per-minute), systolic blood pressure (SBP) (mmHg), diastolic blood pressure (DBP) (mmHg) and BMI (kg/m^2^) (Table 1).

### 2.3. Interventions

The experimental protocols were split into three phases, with a minimum of 48 h between them. The study was completed between 10:00 a.m. and 02:00 p.m. to standardize circadian effects in a soundless room with humidity amid 65% and 70% and temperature between 22 °C and 24 °C [23]. The first day was used to screen partakers through an interview. The eligible subjects were instructed to refrain from mouthwash, drinking alcohol, NO_3_-rich foods and drinks [24], avoid caffeinated beverages or foodstuffs (e.g., coffee, sporting energy drinks and chocolate) and negate exhaustive exercise during the study implementation. Subjects were told to wear comfortable clothing to permit the required physical effort and only eat a light meal two hours prior to the experimental procedures.

On the second day, the subjects ingested a beetroot extract (600 mg capsule) or starch (600 mg placebo capsule) 120 min prior to the experimental procedure; that interval was chosen to allow an acceptable period for digestion, absorption and the performance of any physiological effects [25]. The participants ingested the opposite intervention (placebo or beetroot extract) on the third and final day to guarantee the study’s’ cross-over. On the second and third days, participants undertook physical exercise on a treadmill (Spert ATL, Inbrasport^®^ ATL 2000, Porto Alegre, Brazil) in the first 5 min with HR between 50% and 55% of the estimated maximum HR (208–0.7 × age) [26] for “warm-up”, then afterwards for 20 min with HR equivalent to 65% to 70% of the maximum HR estimated. Last of all, the subjects were seated and monitored for an additional 60 min.

### 2.4. Blinding and Randomization

These opaque capsules were visibly identical; neither the researcher nor the subject could identify the contents of the capsules. An independent researcher who did not participate in the data collection was responsible for selecting the capsules and allocating them to the researcher. The website https://www.randomizer.org/ (accessed on 10 November 2019) was required for the randomization.

The beetroot extract was attained in its commercial form. According to producer (Florien^®^, Sao Paulo, Brazil) information, the active part was derived from its root. Its chemical composition was formulated of: Dry extract (10%) standardized in 10% Betaine and 2.5% NO_3_. Sugars: Sucrose (15–20%), Fructose and Glucose; Mineral salts: Potassium, Sodium, Calcium, Magnesium, Iron (traces); Vitamins: A, B1, B2 and C; Fibers; Glutamine; Pigments: Betanidine and Betaine; Volatile Substances: Pyridine; Rafanol; Saponins; Alkaloid: Betalain; Flavonoids: Isorhamnetin).

### 2.5. Outcomes

#### Blood Pressure

The subjects remained seated during the SBP and DBP measurements, which were attained by auscultatory method with a stethoscope (Littman Classic II, St. Paul, MN, USA) and an aneroid sphygmomanometer (Welch Allyn Tycos, New York, NY, USA) on the subjects’ left arm [27]. SBP and DBP were logged at the subsequent moments: Rest—125th minute after capsule ingestion—and during recovery—1st, 5th, 10th, 20th, 30th, 40th, 50th and 60th minutes after exercise. Mean pulse pressure (PP) was considered as the difference between SBP and DBP (PP = SBP − BP). Mean arterial pressure (MAP) was attained by adding one-third of PP to DBP (MAP = 1/3(PP) + DBP) [27].

### 2.6. HR and HRV Analysis

The HR was achieved beat-to-beat throughout the procedures by an HR monitor (Polar RS800cx, Polar^®^, Kempele, Finland). HRV analysis was completed according to the European Society of Cardiology and the North American Society of Pacing and Electrophysiology guidelines [28].

We nominated a stable series of 256 consecutive RR intervals [29]. The time-domain index of HRV was evaluated by root mean square of successive differences between normal heartbeats (RMSSD) index and frequency-domain index was measured by the high-frequency spectral component (HF) of the power spectral density (0.15 to 0.40 Hz) in absolute units (ms^2^) [13].

The HR and HRV indexes were logged at the following periods: Rest (R1: 120th to 125th min of resting after capsule ingestion) and during recovery: 0 to 5th min; 5th to 10th min; 15th to 20th min; 25th to 30th min; 35th to 40th min; 45th to 50th min and 55th to 60th min (Figure 2).

To compute the HRV indices, we required the Kubios HRV software package (Kubios^®^ HRV version 1.1, University of Kuopio, Kuopio, Finland) [30].

### 2.7. Sample Size

The sample size computation was achieved by a pilot study performed on five subjects. We enforced the online software from the website www.lee.dante.br (accessed on 20 February 2020), which provided the scale of the difference and we computed the root mean square standard deviation of HRV as reference. We measured a 12.7 ms standard deviation and the magnitude of the difference was 13.9 ms. The sample size was a minimum of 11 subjects per group, with an alpha risk of 5% and a beta risk of 80%.

### 2.8. Statistical Analysis

Shapiro-Wilk statistical test was enforced to assess data normality [31]. We applied two-way analysis of variance (ANOVA2) to analyze the differences between time (rest vs. recovery from exercise), the differences between the interventions (placebo vs. beet extract) and, furthermore, to identify a possible interaction effect between them. For repeated measurements, the Greenhouse-Geisser correction was performed for ε < 0.75. The relevant main effects of treatment or interaction effects were subsequently investigated via Ryan-Holm-Bonferroni correction to adjust the results of *p*-value for multiple comparisons. Statistical significance was set at *p* < 0.05 (or, <5%) for all analyses. Cohen’s *d* calculated the effect size to measure the magnitude of changes for significant differences. The confidence interval was considered by a probability of 95% [32]. Assessments were attained via the Statistical Package for the Social Sciences (SPSS) (IBM^®^ SPSS Statistics version 22.0, IBM Corp, Armonk, NY, USA).

## 3. Results

### 3.1. Sample Profile

The characterization (for example; age, body mass, height and BMI (kg/m^2^)) of the sixteen healthy males are illustrated in Table 1.

### 3.2. Blood Pressure following from Exercise

Significant changes were revealed between placebo and beetroot protocols in post-exercise SBP and DBP values vs. Rest (time effect). In the placebo protocol, compared to rest, SBP remained significantly higher for 5 min post-exercise (Rest: 119 ± 8 [95%CI = 115–123] vs. 1th min: 134 ± 8 [95%CI = 130–138] (Cohen’s *d* = 1.73); Rest vs. 5th min: 124 ± 6 [95%CI = 121–127] (Cohen’s *d* = 0.99), *p* < 0.001) of recovery. In the beetroot protocol, compared to rest, SBP established significantly higher values only during 1 min after exercise (Rest: 119 ± 10 [95%CI = 115–124] vs. 1th min: 133 ± 8 [95%CI = 129–136] (Cohen’s *d* = 1.35), *p* < 0.001); Rest vs. 5th min: 123 ± 7 [95 %CI = 119–126], *p* > 0.05], indicating that beetroot extract was able to reduce SBP following exercise more quickly (Table 2).

For DBP, significant differences were achieved compared to rest in the 1st min post-exercise (Rest: 80 ± 3 [95%CI = 78–82] vs. 1th min: 84 ± 10 [95%CI = 80–89] (Cohen’s *d* = 0.56), *p* < 0.001) in the placebo protocol only (Table 2).

In the placebo protocol, MAP values following exercise related to rest continued to be higher for 1 min (Rest: 93 ± 5 [95%CI = 77–109] vs. 1st min: 101 ± 8 [95%CI = 89–113] Cohen’s *d* = 1.22), *p* < 0.001). In the beetroot protocol, the contrast with rest exposed a significant increase in MAP also in the 1th min of recovery from exercise (Rest: 93 ± 6 [95%CI = 80–106] vs. 1th min: 100 ± 7 [95%CI = 87–113] Cohen’s *d* = 1.01, *p* < 0.001). In the recovery analysis, in both the placebo and the beetroot protocol, no significant changes were attained for the PP (Table 2).

No group effect was exposed between the protocols for SBP (*p* = 0.90), DBP (*p* = 0.88), MAP (*p* = 0.73) and PP (*p* = 0.99) and no significant changes (group vs. time) were observed between the values of SBP (*p* = 0.75), DBP (*p* = 0.79), MAP (*p* = 0.93) and PP (*p* = 0.63) between placebo and beetroot protocols.

### 3.3. HR and HRV Recovery after Exercise

Significant changes were recognized between the placebo and beetroot protocols for HRmean and HRV recovery vs. Rest (time effect). Mean HR and HF index had a faster recovery in the beetroot protocol than in the placebo protocol.

Throughout the placebo protocol, the HR mean remained high for 10 min following the physical exercise session (Rest: 76 ± 9 [95%CI = 72–81] vs. 0–5 min: 91 ± 8 [95%CI = 88–95] Cohen’s *d* = 1.75; Rest vs. 5–10 min: 85 ± 9 [95%CI = 81–90] Cohen’s *d* = 0.96, *p* < 0.001). In the beetroot protocol, the HR mean presented an increase after exercise only during 0–5 min recovery (Rest: 78 ± 12 [95%CI = 72–84] vs. 0–5 min: 92 ± 9 [95%CI = 88–97] Cohen’s *d* = 1.26, *p* < 0.001; Rest vs. 5–10 min: 86 ± 9 [95%CI = 82–90], *p* > 0.05) (Figure 3).

In the placebo protocol, in the recovery analysis (Rest vs. recovery) the HF index established a reduction by 10 min after exercise (Rest: 313.25 ± 246.39 [95%CI = 192.51–433.98] vs. 0–5 min: 119.06 ± 96.99 [95%CI = 71.53–166.59] Cohen’s *d* = −1.00; Rest vs. 5–10 min: 120.50 ± 83.53 [95%CI = 79.57−161.42] Cohen’s *d* = −1.01, *p* < 0.001). In the beetroot protocol, HF values remained diminished for just the 5 min after physical exercise (Rest: 331.87 ± 234.82 [95%CI = 216.81–446.93] vs. 0–5 min: 148.50 ± 131.80 [95%CI = 83.91–213.08] Cohen’s *d* = −0.93, *p* < 0.0001; Rest vs. 5–10 min: 160.87 ± 122.30 [95%CI = 100.94–220.80, *p* > 0.05) (Figure 3).

In the placebo protocol, in the recovery analysis (Rest vs. recovery) the RMSSD index exhibited a reduction completely 10 min following exercise (Rest: 31.22 ± 10.44 [95%CI = 26.10–36.34] vs. 0–5 min: 16.91 ± 6.93 [95%CI = 13.51–20.31] Cohen’s *d* = −1.61; Rest vs. 5–10 min: 19.34 ± 6.41 [95%CI = 16.20–22.48] Cohen’s *d* = −1.37, *p* < 0.001). In the beetroot protocol, the changes were uncertain and the RMSSD remained decreased for 10 min after exercise (Rest: 31.52 ± 13.04 [95%CI = 25.13–37.91] vs. 0–5min: 18.16 ± 8.41 [95%CI = 14.04–22.29] Cohen’s *d* = −1.21; Rest vs. 5–10 min: 20.67 ± 8.01 [95%CI = 16.74–24.59] Cohen’s *d* = 1.00, *p* < 0.001 (Figure 3).

No group effect was recognized between the beetroot and placebo protocols on HR mean (*p* = 0.99) and for the HF (*p* = 0.90) and RMSSD (*p* = 0.67) indices. Likewise, no significant changes (group vs. time) were found between the values of HR mean (*p* = 0.60), HF (*p* = 0.69) and RMSSD (*p* = 0.95) between the placebo and beetroot protocols.

## 4. Discussion

Our study identified that nutraceuticals derived from beetroot extract slightly accelerates the recovery of cardiovascular and autonomic parameters in response to exercise, in healthy males via the following observations:(a)In the beetroot protocol immediately after exercise the means of the HR, SBP and DBP values were reduced more quickly.(b)HF indices (typical of vagal modulation) recovered earlier after exercise cessation in the beetroot vs. placebo protocol.

The nutraceuticals’ antioxidant properties from beetroot have been presented as constituents capable of heightening cardiovascular responses and the autonomic response to physical exercise [33]. Thus, based on the central role that NO plays in the homeostasis of the cardiovascular system, the consumption of beetroot extract can source deviations in the baroreflex, elevating HRV and lessening the HR and BP of individuals. Besides, NO_3_ concentrations presented in this study were deficient and thus we cannot conclude that the results occurred owing to the effects of NO_3_.

Earlier studies have established promising results from beetroot compounds on the behavior of vagal HRV indices in response to exercise. Bond et al. [12] presented 500 mL of beetroot juice (approximately 750 mg NO_3_) to 13 African-American females and, after 120 min, subjected them to exercise sessions on a stationary bicycle at 40% and 80% VO2 max intensity. SDNN (Standard deviation of normal to normal R-R-intervals) index values were larger before and during exercise with beetroot juice than throughout the placebo treatment.

These results conform to the effects detected in a study previously published by Benjamim et al. [19]. Twelve healthy male adults were assessed over two days in randomized protocols (beetroot extract 600 mg in the capsule and placebo 600 mg starch in an identical capsule). Then, the subjects remained seated for 120 min at rest, followed by a strength exercise for the lower limbs at an intensity of 75% of 1RM. Next, they were at rest yet again for 60 min. It was likely that in the group that ingested the beetroot extract, there was an acceleration of the recovery of the SDNN, HF and RMSSD indexes.

Notay et al. [11] recognized that the application of 70 mL of NO_3_-rich beetroot juice (6.4 mmol NO_3_^−^) in 14 volunteers (7 women) 165 to 180 min prior to exercise was capable of reducing sympathetic activity before and during physical activity.

In contrast to the formerly cited research studies, we discover controversial results that 600 mg of beetroot extract is able to enhance recovery of vagal heart rhythm modulation. In previous studies, the beetroot extract accelerated the recovery of the SDNN [12] and RMSSD [19]. In our results, only HF index was improved, but differently to RMSSD.

In this sense, beetroot extract appears to be more effective in recovering cardiovascular and autonomic activity after high-intensity exercise. In Benjamim et al. [19], the results were more noticeable than in this study. In their study, the differences between the placebo protocol and the beet protocol was estimated at between 15 to 20 min for recovery of parameters, whilst here it was only by approximately 5 min. We realize that this is by reason of the type of exercise applied (high intensity and short duration) in the Benjamin et al. [19] study, which demanded more substantial cardiac activity. We consider that 65% to 70% of the HRmax is a low intensity for the beet extract to contribute more efficiently to the recovery of cardiac and autonomic parameters after aerobic exercise. Upcoming studies are required to research the beet extract effects at maximum effort to approve these assumptions.

The improvement in these indices specifies a quicker reactivation of the vagus nerve in the post-exercise period in the beetroot extract protocol compared to the placebo. Confirmation is required that a slow post-exercise autonomic recovery, analyzed by HRV indices that assess vagal modulation, is related to increased cardiovascular risk [34].

To provide corresponding information with regards to the influence of beetroot extract on the ANS, we also estimated the hemodynamic parameters of BP and HR as secondary outcomes. Before, it was illustrated that beetroot extract recovered HR faster in the post-exercise period [19]. The results of this study strengthened this. Yet Bond et al. [12] could not observe any influence of beetroot juice on HR. The HR recovery in the post-exercise period is influenced by the reactivation of the parasympathetic nervous system. Considering that the decrease in vagal activity after exercise is linked with the risk of mortality, this permitted the HR recovery after a bout of exercise to be a resolute predictor of mortality [14].

In the hemodynamic parameters of BP, we revealed that the beetroot extract accelerated slightly the recovery of SBP and DBP values compared to placebo. Previous studies corroborate our findings, while more sensitive results are found during high-intensity exercise [19].

In the study by Carrijo et al. [35], the effects of different NO_3_ concentrations in beetroot juice on the HRV of hypertensive postmenopausal women were compared. HRV was assessed for 20 min after sitting at rest, 120 min after drinking one of the drinks and after performing 40 min of aerobic exercise at 65% and 70% of HR reserve on a treadmill. For the subsequent analysis, HRV was logged for 90 min after exercise for time, frequency and non-linear domains.

These researchers described non-significant effects of beetroot juice on HRV indexes, HR [34] and BP [35]. Ostensibly, this is evidence contrary to that presented in the research literature [2,3,4,5,18]. So far, the low amount of NO_3_ provided (approximately 200 mg) in the study may contribute to the results of Carrijo et al. [35]. The researchers vindicated that, despite a probable increase in NO_3_ bioavailability, fluctuations in autonomic function were only induced by exercise and this result is well-matched with our results.

Another reason for the lack of influence of beetroot juice is that there may have been a clearance of NO_3_ and nitrite during the exercise itself [35]. Yet we cannot be certain, as data on NO_3_ and NO_2_ concentrations after exercise were not presented by Carrijo et al. [35].

Differences between studies seem segmented by exercise intensity and these conflicts may be explained by the intensity of exercise attributed to the methodology. Cardiovascular and autonomic responses in continuous vs. high-intensity exercise are different. In continuous exercise, the HR, BP and sympathetic flow are increased graduate following o time-execution. In high-intensity exercise, these responses are directed towards a peak, but between rest and re-test the increases in HR, BP and decreases in HRV are more prominent [36]. On the other hand, at the end of the exercise, generally the high-intensity interval requires a longer time until the withdrawal of meta-bore-flex and removal of muscle metabolites [37]. Since these questions have a relationship with the re-establishment of parasympathetic modulation to the heart after exercise [38], the beetroot compounds can be more effective in removing metabolites in high-intensity exercise [39].

These questions should be confirmed in further studies with beetroot compounds being tested in different types of exercise in the same trial to evaluate the cardiovascular and autonomic responses in the same sample.

Our results propose some key areas of interest for clinical and sports nutrition studies, contribute to the health professionals’ performance and reveal novel alternate therapies and interventions.

Our study established aerobic exercise and physically challenged subjects to test beetroot extract. We encourage other studies to replicate our experiments, as we appreciate that just as important when presenting unprecedented results is reassessing the reproducibility of results in the identical population and during additional situations.

Based on the data presented, while nutraceuticals from beetroot extract establish negligible interactions in cardiovascular health, further evidence is necessary so as not to exclude the opportunity that in future this intervention may be applied with other dietary interventions to improve cardiovascular health (e.g., magnesium, vitamin C, catechin-rich beverages, or soy isoflavones) [40,41].

Our study has some strong points as the interventions were randomized and the participants and researchers blinded. Whilst the sample number was small, it exceeded the calculated sample size.

In this study, the oxidative stress biomarker concentrations, besides the concentrations of nitrite/nitrate in plasma, were not measured and were considered a limitation. Further studies with nutraceuticals from beetroot extract must measure biomarkers (e.g., interleukins, inflammatory markers) and plasma NO_2_ and NO_3_ concentrations to comprehend exactly which biological and molecular aspects are related to those effects attained in this study and in other analogous studies [19].

Some points require consideration in our study design. The prescription of submaximal aerobic exercise was based on the HR values and, therefore, this is an important limitation due to the possibly great intervention between subjects included. Although we asked participants to consume the same meal before the experiments, the total NO_3_ content in these meals were not controlled.

There is a limitation for nitrate oral bioconversion, as we administered a capsule. Nevertheless, the capsule is part of the randomized double-blind placebo design. This is one of the most appropriate and viable interventions to evaluate the effect of other compounds, but not foods/extracts containing NO_3_/NO_2_. We highlight that complementary research with clinical populations must be enforced before these effects are considered. At the outset of the study, we had difficulties locating subjects with a BMI < 25 kg/m^2^ and so we redefined our criteria to values up to 29.9 kg/m^2^. We appreciate that this is a limitation. Yet almost half of the world’s population will be overweight in the near future, which will undoubtedly increase the external validity of these results. Studies with female participants are also encouraged. We excluded females on account of the difficulty of encouraging women to participate and problems controlling the menstrual period and its interference with HRV.

## 5. Conclusions

In healthy males, beetroot extract ingestion has negligible effects regarding improvement in the recovery of the cardiovascular and autonomic systems after submaximal aerobic exercise.

## Figures and Tables

**Figure 1 ijerph-20-04019-f001:**
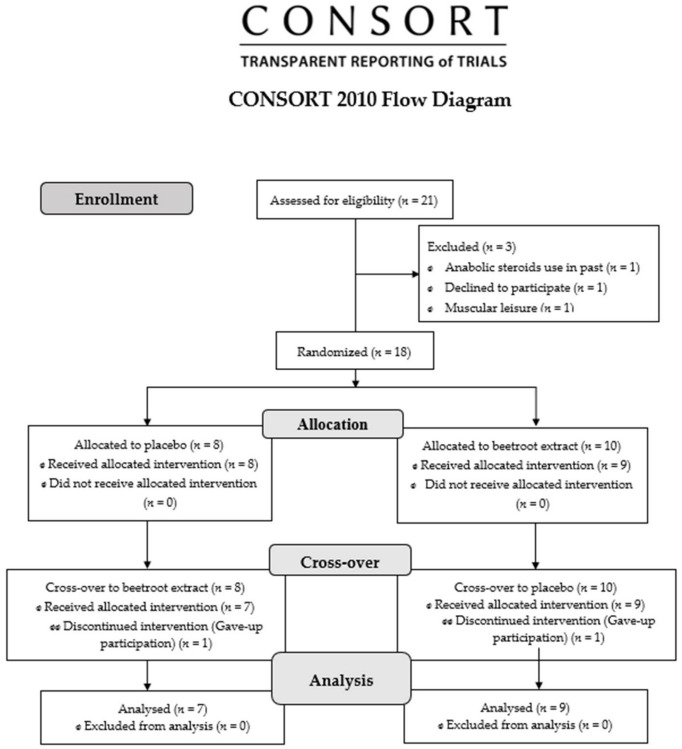
Flow diagram.

**Figure 2 ijerph-20-04019-f002:**
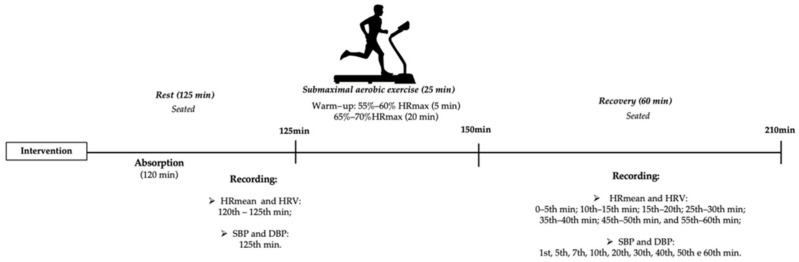
Study design.

**Figure 3 ijerph-20-04019-f003:**
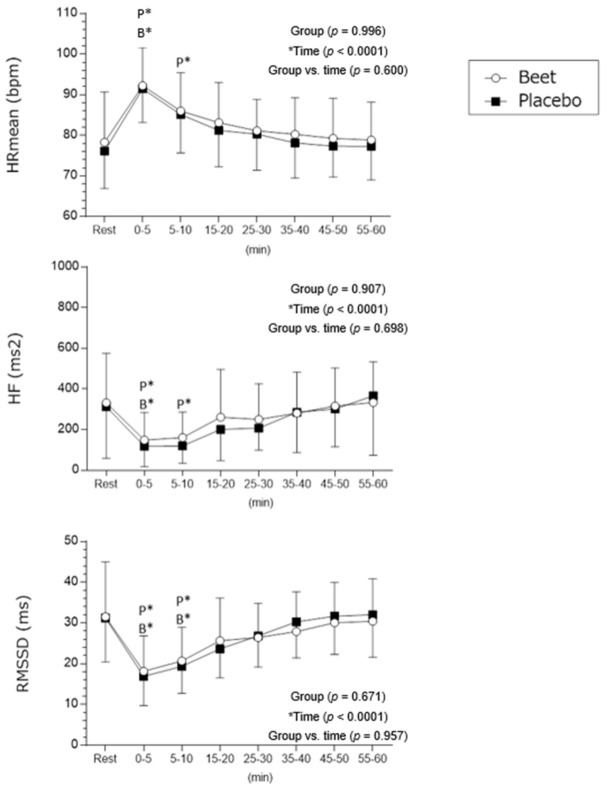
Mean values and respective standard deviations of HR, HF and RMSSD (ms^2^) obtained at rest and during recovery from submaximal aerobic exercise. * Values with significant differences in relation to rest (time effect) (*p* < 0.05) (ANOVA2, Bonferroni’s post-test); RMMSD: Root mean squared of successive RR intervals differences; High frequency (HF) spectrum (0.15 to 0.40 Hz).

**Table 1 ijerph-20-04019-t001:** Mean values and their respective standard deviations (minimum and maximum) of age, mass, height, BMI, heart rate, SBP and DBP.

Variables	Values
Age (years)	21.31 ± 2.26 (18–27)
BMI (kg/m^2^)	24.82 ± 1.99 (20.93–28.72)
Height (m)	173.8 ± 0.06 (1.65–1.87)
Mass (kg)	75.10 ± 7.43 (57–84)
Heart rate (bpm)	74 ± 11 (51–91)
SBP (mmHg)	117 ± 3(110–120)
DBP (mmHg)	77 ± 3 (70–80)

BMI: body mass index; kg: kilogram; m: meters; bpm: beats per minute; mmHg: millimeter of mercury.

**Table 2 ijerph-20-04019-t002:** Systolic and diastolic blood pressure values followed by standard deviations [confidence interval 95%] before and post-exercise in individuals during experimental protocols.

Variables	Treatment	Rest	Rec (1 min)	Rec (5 min)	Rec (10 min)	Rec (20 min)	Rec (30 min)	Rec (40 min)	Rec (50 min)	Rec (60 min)
SBP (mmHg)	Beet	119 ± 9 (114–124)	132 ± 8 * (128–136)	122 ± 7 (119–126)	119 ± 7 (116–122)	119 ± 7 (116–122)	118 ± 6 (115–121)	117 ± 7 (114–120)	117 ± 7 (114–120)	116 ± 6 (113–120)
	Placebo	119 ± 8 (115–123)	134 ± 8 * (129–138)	124 ± 6 * (121–127)	121 ± 6 (117–123)	118 ± 5 (115–120)	118 ± 5 (115–120)	118 ± 5 (115–120)	118 ± 5 (115–120)	118 ± 5 (115–120)
DBP (mmHg)	Beet	80 ± 6 (77–83)	83 ± 7 (79–87)	81 ± 6 (78–85)	82 ± 8 (78–87)	81 ± 7 (78–85)	82 ± 7 (79–86)	81 ± 7 (78–85)	81 ± 7 (77–85)	81 ± 7 (78–85)
	Placebo	80 ± 4 (78–82)	84 ± 10 * (80––89)	83 ± 4 (81–85)	82 ± 4 (80–84)	80 ± 4 (79–83)	80 ± 4 (79– 83)	80 ± 5 (78–82)	80 ± 5 (78–82)	81 ± 3 (80–82)
MAP (mmHg)	Beet	93 ± 6 (80–105)	100 ± 7 * (87–113)	95 ± 5 (81–109)	94 ± 5 (80–108)	94 ± 5 (80–108)	94 ± 5 (80–108)	93 ± 5 (79–108)	93 ± 6 (80–106)	94 ± 6 (80–106)
	Placebo	93 ± 4 (77–109)	101 ± 8 * (89–113)	96 ± 3 (78–114)	95 ± 3 (77–113)	93 ± 3 (75–111)	93 ± 3 (75–111)	92 ± 3 (74–110)	92 ± 3 (74–110)	93 ± 3 (71–115)
PP (mmHg)	Beet	39 ± 8 (28–5)	48 ± 8 (37–59)	40 ± 7 (28–52)	36 ± 11 (27–46)	37 ± 9 (27–47)	36 ± 9 (25–46)	35 ± 7 (23–47)	36 ± 6 (23–49)	35 ± 7 (23–46)
	Placebo	39 ± 6 (26–52)	49 ± 11 (39–58)	41 ± 7 (29–52)	38 ± 7 (26–50)	38 ± 7 (24–50)	37 ± 6 (24–50)	38 ± 7 (26–50)	38 ± 6 (26–49)	36 ± 6 (22–50)

Rest: pre-exercise; Rec: Recovery post-exercise; SBP: systolic blood pressure; DBP: diastolic blood pressure; mmHg: millimeter of mercury. * Values with significant differences in relation to rest (time effect) (*p* < 0.05) (ANOVA2, Bonferroni’s post-test).

## Data Availability

The data presented in this study are available on request from the corresponding author.

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
