# Peer review of "Negligible Effects of Nutraceuticals from Beetroot Extract on Cardiovascular and Autonomic Recovery Response following Submaximal Aerobic Exercise in Physically Active Healthy Males: A Randomized Trial"

_ijerph, 2023, doi:10.3390/ijerph20054019_

Round 1
Reviewer 1 Report
I thank the authors for their interesting study investigating the influence of beetroot extract on the recovery of BP and HR following submaximal treadmill running.
The study design (crossover RCT) is appropriate. However, the statistical methods used are a major flaw that need to be addressed before it is reviewed in further detail. I believe a two-way ANOVA with repeated measures needs to be used to examine whether there is an interaction between the 2 factors (treatment (beetroot/placebo) and time) on the dependent variables. Please can the authors seek statistical assistance and revise their analyses to ensure their data are adequately analysed.
Without proper statistical analysis, the results and their interpretation are not valid.
A second major problem is the study design is different from the pre-registered protocol. The pre-registered protocol method uses resistance exercise, not aerobic exercise. https://clinicaltrials.gov/ct2/show/NCT04094233
Please see below general comments that need to be addressed please:
The abstract does not include any data or P values.
The CONSORT diagram does not include the inclusion/exclusion criteria described in the text that was used to screen participants. This detail needs to be included. Reasons for not receiving the intervention and discontinuing the intervention need to be described please.
I note there are discrepancies between the CONSORT diagram in the manuscript and those in a pre-print available online which is a concern. Please can the authors account for this.
In table 1 HR and BP are reported to 2 decimal places. Were these measured to this level of precision? If so, how was this done? If not, please revise.
That the nitrate content of the pre-exercise meals was not controlled, and may have been different between trials, needs to be discussed as a limitation.
That exercise was prescribed using an estimate of HR max should be discussed as a possible limitation. Comment on the validity of the estimation equation used.
The hypothesis appears to contrast with the background literature presented, re the rationale for conducting the study. Was the hypothesis a priori? I note there is no hypothesis in the pre-print.
There are numerous grammatical errors. Please can the author who reviewed English grammar and spelling do so in detail or seek assistance with this.
Author Response
We appreciate your substantial and careful revision. We have revised the manuscript to improve it based on your comments. The added or modified words, phrases, and sentences are in red.Reviewer 1
I thank the authors for their interesting study investigating the influence of beetroot extract on the recovery of BP and HR following submaximal treadmill running.
The study design (crossover RCT) is appropriate. However, the statistical methods used are a major flaw that need to be addressed before it is reviewed in further detail. I believe a two-way ANOVA with repeated measures needs to be used to examine whether there is an interaction between the 2 factors (treatment (beetroot/placebo) and time) on the dependent variables. Please can the authors seek statistical assistance and revise their analyses to ensure their data are adequately analysed.
Without proper statistical analysis, the results and their interpretation are not valid.
ANSWER: We thank you for this consideration. We have assessed these questions to resolve doubts about our results. A priori, we believe that the ANOVA-one-way is appropriate for your main objectives as it analyzes the recovery of HRV, HR, and BP after exercise. However, we understood that comparisons between interventions (group effects and interaction effects) are essential to improve the interpretation of these results.
A second major problem is the study design is different from the pre-registered protocol. The pre-registered protocol method uses resistance exercise, not aerobic exercise. https://clinicaltrials.gov/ct2/show/NCT04094233
ANSWER: We thank you for your essential consideration. We inserted the correct register protocol number (https://clinicaltrials.gov/ct2/show/NCT05294198) registered in May of 2022.
Please see below general comments that need to be addressed please:
The abstract does not include any data or P values.
ANSWER: We thank you for this commentary. We inserted the P values in the abstract.
The CONSORT diagram does not include the inclusion/exclusion criteria described in the text that was used to screen participants. This detail needs to be included. Reasons for not receiving the intervention and discontinuing the intervention need to be described please.
I note there are discrepancies between the CONSORT diagram in the manuscript and those in a pre-print available online which is a concern. Please can the authors account for this.
ANSWER: We thank you for this critical question. We changed the information provided in the CONSORT diagram. During the submission process, we sent the CONSORT diagram that does not actualize as was in the pre-print.
In table 1 HR and BP are reported to 2 decimal places. Were these measured to this level of precision? If so, how was this done? If not, please revise.
ANSWER: We thank you for this vital commentary. We transformed these decimal places to usually clinical values.
That the nitrate content of the pre-exercise meals was not controlled, and may have been different between trials, needs to be discussed as a limitation.
That exercise was prescribed using an estimate of HR max should be discussed as a possible limitation. Comment on the validity of the estimation equation used.
ANSWER: We agree that these questions are significant limitations of our methodology. These questions are imputed in the discussion section.
The hypothesis appears to contrast with the background literature presented, re the rationale for conducting the study. Was the hypothesis a priori? I note there is no hypothesis in the pre-print.
ANSWER: We thank you for this critical evaluation. Although the literature shows that beetroot compounds can help the cardiovascular system, the literature body refers to high-intensity and cut-duration exercises. Our initial hypothesis for this study is that beetroot compounds would not be able to evoke improvement in cardiovascular responses.
There are numerous grammatical errors. Please can the author who reviewed English grammar and spelling do so in detail or seek assistance with this.
ANSWER: We thank you for this critical evaluation. The Native speaker co-author revised the manuscript in detail to improve the English grammar and spelling as required.

Reviewer 2 Report
· This study has a Good Title and the results are useful.
· The analysis method is suitable.
· It has Correct referencing and citations.
· There is Correct and proper English writing
· My decision about this manuscript is accepted.
Author Response
We really appreciate your positive evaluation about our manuscript.
Reviewer 3 Report
This paper explored the effects of beetroot extract on the recovery of cardiovascular and autonomic parameters after a submaximal aerobic exercise session, providing experimental basis for the specific application of beetroot extract in exercise practice. However, there are the following problems:
(1) HRV detection includes time-domain index (SDNN, SDANN, RMSSD) of 24-hour and frequency domain index (HF, LF and LH/HF values) of 5-minute static supine position records. RMSSD and HF are selected in this paper to reflect HRV, however, the basis and significance of their selection are not found in the introduction or discussion.
(2) In the materials and methods, some descriptions (such as P190-P193) are inconsistent with the contents in Figure 2.
(3) In the experimental design, the index values measured from the 90th to 95th min after taking beetroot extract were taken as the baseline values, and then 25 minutes of aerobic exercise (including 5 minutes of warm-up) were carried out, and the blood pressure were measured at the 1st, 5th, 10th, 20th, 30th, 40th, 50th, and 60th minutes after exercise; and HRV were measured in 0th to 5th min, 5th to 10th min, 15th to 20th min, 25th to 30th min, 35th to 40th min, 45th to 50th min, and 55th to 60th min to explore the effect of beetroot extract on the recovery of cardiovascular and autonomic parameters after submaximal aerobic exercise.. Therefore, we can take the blood pressure value at the first minute and the HRV value from 0th to 5th min after exercise as the values immediately after exercise, and compare the values immediately after exercise with the rest value before exercise to reflect the effect of the supplement of beetroot extract on the cardiovascular and autonomic regulation during exercise, using the blood pressure at the 5th, 10th, 20th, 30th, 40th, 50th, and 60th minutes after exercise and the HRV value from 5th to 10th min, 15th to 20th min, 25th to 30th min, 35th to 40th min, 45th to 50th min, and 55th to 60th min were compared with the value immediately after exercise to reflect the effect of beetroot extract on the recovery of cardiovascular and autonomic parameters after exercise. Therefore, if the experimental results can be compared in sections, it can not only explore the effects of the supplement of beetroot extract on cardiovascular and autonomic regulation during exercise, but also explore the effects of beetroot extract on the recovery of cardiovascular and autonomic parameters after exercise. Of course, if we can supplement the values of blood pressure and HRV before taking beetroot extract, we can better to explore the effects of beetroot extract on cardiovascular and autonomic regulation at rest.
(4) In the discussion, although the author can analyze the research results according to the corresponding literature, the depth of discussion and analysis is not enough. According to the experimental results, the characteristics of blood pressure and HRV changes during high-intensity and sub-maximum intensity exercise and the corresponding physiological mechanism should be analyzed in detail. On this basis, the possible mechanism of the supplement of beetroot extract on the regulation of blood pressure and HRV after different intensities of exercise should be analyzed.
(5) Some words in the paper are misspelled, such as it were (p331), reseachers (P361), etc.
(6) Some references are labeled incorrectly, for example, Benjamim et al. 18 (P314),which should be carefully checked.
Author Response
REVIEWER 3
We appreciate your important and careful revision. We have revised the manuscript to improve it based on your comments. The added or modified words, phrases, and sentences are in red.
This paper explored the effects of beetroot extract on the recovery of cardiovascular and autonomic parameters after a submaximal aerobic exercise session, providing experimental basis for the specific application of beetroot extract in exercise practice. However, there are the following problems:
(1) HRV detection includes time-domain index (SDNN, SDANN, RMSSD) of 24-hour and frequency domain index (HF, LF and LH/HF values) of 5-minute static supine position records. RMSSD and HF are selected in this paper to reflect HRV, however, the basis and significance of their selection are not found in the introduction or discussion.
ANSWER: We thank you for this essential commentary. We inserted these requirements in the introduction section with RMSSD and HF indexes representation.
2nd paragraph: “In the past few years, some scientific publications have applied the RMSSD (root mean square standard deviation) and HF (high frequency) indexes frequently as a way to assess the cardiac vagal modulation recovery in response to physical exercise. These two indexes have been revealed to be fairly accurate to comprehend the cardiac autonomic modulation. The RMSSD index is highly representative of the parasympathetic modulation of the heart, and the HF index experiences the negative influence of respiratory frequency13”
(2) In the materials and methods, some descriptions (such as P190-P193) are inconsistent with the contents in Figure 2.
ANSWER: We thank for that essential observation. The intervention was intake 120 minutes before any data collection. We corrected this information in Figure 2 as well in methods description.
(3) In the experimental design, the index values measured from the 90th to 95th min after taking beetroot extract were taken as the baseline values, and then 25 minutes of aerobic exercise (including 5 minutes of warm-up) were carried out, and the blood pressure were measured at the 1st, 5th, 10th, 20th, 30th, 40th, 50th, and 60th minutes after exercise; and HRV were measured in 0th to 5th min, 5th to 10th min, 15th to 20th min, 25th to 30th min, 35th to 40th min, 45th to 50th min, and 55th to 60th min to explore the effect of beetroot extract on the recovery of cardiovascular and autonomic parameters after submaximal aerobic exercise.. Therefore, we can take the blood pressure value at the first minute and the HRV value from 0th to 5th min after exercise as the values immediately after exercise, and compare the values immediately after exercise with the rest value before exercise to reflect the effect of the supplement of beetroot extract on the cardiovascular and autonomic regulation during exercise, using the blood pressure at the 5th, 10th, 20th, 30th, 40th, 50th, and 60th minutes after exercise and the HRV value from 5th to 10th min, 15th to 20th min, 25th to 30th min, 35th to 40th min, 45th to 50th min, and 55th to 60th min were compared with the value immediately after exercise to reflect the effect of beetroot extract on the recovery of cardiovascular and autonomic parameters after exercise. Therefore, if the experimental results can be compared in sections, it can not only explore the effects of the supplement of beetroot extract on cardiovascular and autonomic regulation during exercise, but also explore the effects of beetroot extract on the recovery of cardiovascular and autonomic parameters after exercise. Of course, if we can supplement the values of blood pressure and HRV before taking beetroot extract, we can better to explore the effects of beetroot extract on cardiovascular and autonomic regulation at rest.
ANSWER: We thank you for this important evaluation. We consider that the better scenario to evaluate the beetroot extract would be to made measurements before any intervention (extract or placebo). However, due the unavailability of participants to spend a long time in lab (add 120 minutes before any measurements) we did cannot to performed the experiments in this way. Now, through analysis with ANOVA-two-way we didn`t identified any significant different in baseline between groups. This reinforce that beetroot extract was unable to evokes changes in cardiovascular/autonomic parameters before exercise. Furthermore, the net results between protocols at the REST-time are really negligible.
(4) In the discussion, although the author can analyze the research results according to the corresponding literature, the depth of discussion and analysis is not enough. According to the experimental results, the characteristics of blood pressure and HRV changes during high-intensity and sub-maximum intensity exercise and the corresponding physiological mechanism should be analyzed in detail. On this basis, the possible mechanism of the supplement of beetroot extract on the regulation of blood pressure and HRV after different intensities of exercise should be analyzed.
ANSWER: We thank you for your essential commentary. We add a paragraph in the discussion regarding these questions.
Paragraph: “Differences between studies seem segmented by exercise intensity, and these conflicts may be explained by the intensity of exercise attributed to the methodology. Cardiovascular and autonomic responses in continuous vs. high-intensity exercise are different. In continuous exercise, the HR, BP, and sympathetic flow are increased graduate following o time-execution. In high-intensity exercise, these responses have behavior direction to peak, but between rest and re-test, the increases in HR, BP, and drops in HRV are more prominent (Price et al. 2020). On the other hand, at the end of the exercise, generally, the high-intensity interval requires a longer time until the withdrawal of metaboreflex and muscle metabolites remove (Thomas et al. 2004). Since these questions have a relationship with the re-establishment of parasympathetic modulation to the heart after exercise (Pecanha et al. 2016), the beetroot compounds can be more effective in removing metabolites in high-intensity exercise (Sperling et al. 2016).”
(5) Some words in the paper are misspelled, such as it were (p331), reseachers (P361), etc.
(6) Some references are labeled incorrectly, for example, Benjamim et al. 18 (P314),which should be carefully checked.
ANSWER: We thank for these appointments. We corrected the English review spell and grammar, furthermore, a English native (co-author) revised the whole article to improve vocabulary and any suitable misspelling.

Round 2
Reviewer 3 Report
Although the author has revised the paper, the following problems still exist:
(1) The description of the statistical method seems to be inaccurate. In this paper, “the two-way analysis of variance (ANOVA2) (group x time x time point) ” is used. What does “time” represent? What does “time point” mean? From the perspective of “group x time x time point”, it seems to be three factors. Taking SBP as an example, I suggest to analyze the SBP value of Rest and Rec (1min) in Beet group and Placebo group by double-factor analysis of variance of repeated measurements (group x exercise) to reflect that beetroot extract on the reaction degree of SBP during acute exercise. Then, the SBP value of Rec (1min), Rec (5min), Rec (10min), Rec (20min), Rec (30min), Rec (40min), Rec (50min), Rec (60min) in the Beet group and Placebo group were subjected to a two-factor analysis of variance for repeated measurements (group x time) to reflect that beetroot extract on the recovery of SBP following acute exercise. Then modify the description of the experimental results.
(2)“Although these results to be identified in time-analysis in comparison to rest, there`s no difference among the values of cardiovascular and autonomic variables analyzed in this study.” (line 198-200)
The statement of this sentence seems to be inconsistent with the research results. From the research results, the values of cardiovascular and autonomic variables have time effects, especially 0-5 min and 5-10 min following exercise, but these values have no inter-group effect. I hope you can proofread it carefully.
(3) Further optimize the discussion, and unify the dimension format of references according to the requirements of this journal, especially the supplementary content.
Author Response
We appreciate your substantial and careful revision. We have revised the manuscript to improve it based on your comments. The added or modified words, phrases, and sentences are in red.
REVIEWER: Although the author has revised the paper, the following problems still exist:
The description of the statistical method seems to be inaccurate. In this paper, “the two-way analysis of variance (ANOVA2) (group x time x time point) ” is used. What does “time” represent? What does “time point” mean? From the perspective of “group x time x time point”, it seems to be three factors. Taking SBP as an example, I suggest to analyze the SBP value of Rest and Rec (1min) in Beet group and Placebo group by double-factor analysis of variance of repeated measurements (group x exercise) to reflect that beetroot extract on the reaction degree of SBP during acute exercise. Then, the SBP value of Rec (1min), Rec (5min), Rec (10min), Rec (20min), Rec (30min), Rec (40min), Rec (50min), Rec (60min) in the Beet group and Placebo group were subjected to a two-factor analysis of variance for repeated measurements (group x time) to reflect that beetroot extract on the recovery of SBP following acute exercise. Then modify the description of the experimental results.
ANSWER: We thank for your careful evaluation about this point and we understand that previous description may be confuse. Indeed, we changed the description of statistical analysis to avoid misunderstanding about how the analysis were performed.
“We applied two-way analysis of variance (ANOVA2) to analyse the differences between time (rest vs. recovery from exercise) and the differences between the interventions (placebo vs. beet extract), furthermore, to identify a possible interaction effect between them”.
REVIEWER: “Although these results to be identified in time-analysis in comparison to rest, there`s no difference among the values of cardiovascular and autonomic variables analyzed in this study.” (line 198-200). The statement of this sentence seems to be inconsistent with the research results. From the research results, the values of cardiovascular and autonomic variables have time effects, especially 0-5 min and 5-10 min following exercise, but these values have no inter-group effect. I hope you can proofread it carefully.
ANSWER: Thank for your important commentary. We tried discuss exactly that the differences were achieved with recovery analysis (time analysis – comparing rest values vs. exercise and recovery values) but not when intergroup (time x group) analysis were performed. To avoid further doubts and misunderstanding about that, we preferred delete this mention from the manuscript.
REVIEWER: Further optimize the discussion, and unify the dimension format of references according to the requirements of this journal, especially the supplementary content.
ANSWER: Thank for your important consideration. We reviewed the manuscript in accordance to the requirements of the journal.
